# A Novel Blockchain-Enabled Supply-Chain Management Framework for Xinjiang Jujube: Research on Optimized Blockchain Considering Private Transactions

**DOI:** 10.3390/foods12030587

**Published:** 2023-01-30

**Authors:** Hao Song, Wenfei Ge, Pan Gao, Wei Xu

**Affiliations:** 1College of Information Science and Technology, Shihezi University, Shihezi 832003, China; 2College of Agriculture, Shihezi University, Shihezi 832003, China

**Keywords:** blockchain, jujube supply chain, network optimization, transaction-workflow division, hyperledger

## Abstract

Excellent jujube supply-chain management is of great significance to the development of the Xinjiang jujube industry. However, traditional jujube supply-chain management is faced with the dilemmas of opaque connection of transaction flow and capital flow information, unreliable product traceability information, and jujube farmers’ lack of bargaining power. In this research, we propose a jujube supply-chain-management framework based on blockchain. Hyperledger fabric is the distributed solution platform of this research. The current blockchain-based traceability framework for agri-food emphasizes the construction process and ignores the performance and characteristics of the framework. This research optimizes the blockchain-network-topology architecture and storage cost of writing data. This not only solves the traceability of jujube, but also fills the gap in previous research on the traceability framework. Moreover, transactions are innovatively divided into common transactions and private transactions. Private transactions have enhanced the bargaining power of jujube farmers. Through the analysis of benchmark tests, the effectiveness and feasibility of the framework are verified.

## 1. Introduction

Xinjiang jujube’s supply-chain management (SCM) is relatively backward. As the scale of the supply chain increases and the number of supply-chain stakeholders grows, traditional SCM tends to face multiple dilemmas. First, in an untrustworthy environment, the information docking rules of upstream and downstream stakeholders are unclear. This leads to a low information-sharing rate (ISR), which easily causes bullwhip effect. The highly unbalanced ISR of the whole supply-chain docking link may lead to a reverse bullwhip effect [1]. Second, centralized SCM lacks data transparency, where stakeholders manage their data alone, and tampering costs are very low. Under the traditional supply-chain framework, it is difficult for regulators and consumers to trace the root causes of product quality problems [2]. Furthermore, data are stored in a regular database controlled by SCM service providers. Such a centralized storage model has the potential of single-point failure [3]. Similarly, agri-food SCM, as a subdivision of SCM, also faces the above difficulties. The interdependence of stakeholders in the agri-food supply chain makes the SCM very complex. Establishing a collaborative model in the agri-food supply chain can effectively support food safety and traceability to meet sustainable global challenges [4]. 

The supply chain’s traceability helps solve product recall and counterfeiting issues [5]. For Xinjiang jujube, it is necessary to connect all supply-chain nodes to reduce the time-consuming information exchange between nodes. Agri-food SCM has high requirements for docking information flow, capital flow, and other data upstream and downstream of the supply chain. Radio frequency identification (RFID) technology improves the transparency and efficiency of data exchange with its data recorder function and the introduction of integrated components such as integrated sensors. The disadvantage of RFID technology is that some ultra high frequency (UHF) tags have more interference than other frequencies, and some bands are not allowed to be used globally [6]. Another disadvantage lies in its high implementation cost [7]. A SCM system based on RFID technology stores data in a centralized database, which obviously cannot overcome the problem of single-point failure. Blockchain is based on distributed technology that uses cryptography technology to guarantee data is immutable [8], and smart contract is a pre-defined set of rules which run synchronously on blockchain [9,10]. The decentralized nature of blockchain can well avoid the single-point failure of data storage. Blockchain is a chained data structure that combines data blocks and information blocks in chronological order [11]. Its data structure and cryptography technology make it naturally fit traceability application scenarios. As a transaction-driven state machine, smart contract provides a handle for external interaction with the blockchain ledger. In recent years, researchers, to solve traditional agri-food SCM shortcomings, have explored the use of blockchain technology to reconstruct the conventional agri-food supply chain system.

Many researchers are dedicated to transforming the agri-food SCM with blockchain technology and put forward corresponding analysis methods, frameworks, and practical models. However, current research highlights the characteristics that the agri-food information is difficult to tamper with, and the traceability information is easy to trace and does not take advantage of other merits of blockchain. The framework of multiple chains or multiple logical ledgers reduces the efficiency of information sharing and increases the cost of supervision. This study uses the idea of layering for reference and stores the traceability data in categories. The hash function is used to compress large data blocks onto the chain, and the original data are stored off the chain to relieve the storage pressure on the chain. Thus, a single chain with a clear structure is constructed. While meeting the traceability, the single-chain structure is easy to manage and supervise, and the information sharing is smooth. When we investigated in Alar (Xinjiang, China), we found that local farmers’ jujubes were mainly purchased by cooperatives, and farmers lacked bargaining power in the traditional supply-chain system. This study recognizes that it is necessary to consider the information-interaction channel of the supply-chain framework. By splitting different transaction workflows, farmers’ capital flow information is hidden from middlemen. In this way, middlemen can be prevented from using their monopoly position to formulate procurement strategies to infringe upon the interests of farmers. Given the above research gaps, this paper proposes a novel jujube SCM framework based on hyperledger fabric. As a consortium-blockchain platform, hyperledger fabric has the advantages of inherent modularity and pluggability [12]. Hyperledger Caliper serves as a supporting benchmark tool. It can evaluate the performance of hyperledger-based systems for throughput, latency, transaction rate, memory consumption, CPU utilization, and disc read/write operations [13]. Through the research, we hope to give enlightenment to future research of agri-food SCM. Our main contributions are summarized as follows:(1)We build a Xinjiang jujube SCM framework, which compresses the data written into the blockchain while ensuring traceability to meet the sustainability challenge;(2)We classify the transactions in the supply chain workflow and introduce private data into the framework to improve farmers’ bargaining power;(3)We evaluate the effectiveness of our framework through the analysis of performance metrics obtained from the benchmark.

The remainder of this paper is organized as follows. Section 2 is a brief review of related literature. We present the design and workflow of the Xinjiang jujube SCM framework in Section 3. The design and implementation of key smart contracts are illustrated in Section 4. In Section 5, we present performance analysis and provide illustrative numerical results by benchmark. Finally, we conclude this paper and discuss future work in Section 6.

## 2. Literature Review

Our literature review was carried out from three aspects: RFID-technology adoption in the food/argi-food supply chain, methods for evaluation of building blockchain-based supply-chain systems, and blockchain solutions for the specific agri-food supply chain.

### 2.1. RFID Technology Adoption in Food/Argi-Food Supply Chain

In the SCM system that pursues traceability, a system based on Internet of Things (IoT) technology is the focus of researchers under centralized mode architecture [14]. IoT automatically generates data and transmits them through the Internet by utilizing intelligent devices with sensing and communication capabilities [15]. The electronic tag stores information in digital format, and then the contents of the electronic tag can be read by a particular small integrated circuit scanner. The most commonly used is RFID technology [16]. RFID has the characteristics of mobility and inventory accuracy [17]. It does not need physical contact in the reading stage and is fully automated [18]. Therefore, RFID technology is suitable for offering field control or real-time control during cargo transportation [19]. These advantages enable it to enhance the traceability potential in SCM.

Barge et al. [20] adopted different RFID technologies on different meat products and conducted a benchmark evaluation. This research brings enlightenment to improve the coupling of physical objects and information in the traceability system of agri-food. Grunow and Piramuthu [21] exploited RFID to help reduce waste in the perishable food supply chain. According to Biswal et al., the adoption of RFID in the non-profit supply-chain scenario can reduce storage cost and positively impact the Indian food-safety system [22]. Several studies have shown that RFID technology can improve the sustainability and traceability of agri-food SCM and reduce management costs. However, RFID technology still inevitably stores data in a centralized database. The cost of data tampering is low, whereas the cost of data docking in the agri-food supply chain is high. It is necessary to adopt new technology to overcome the above drawbacks. In Table 1, we compare the early traceability platform, the traceability platform based on RFID technology and the traceability platform based on blockchain technology in multiple dimensions. This reflects the advantages of adopting blockchain technology to reconstruct the traceability platform.

The number 1 in the table represents the early traceability platform, number 2 represents the traceability platform based on RFID technology, and number 3 represents the traceability platform based on blockchain technology. It can be seen that the traceability platform using blockchain technology lags behind the other two architectures only in terms of storage cost.

### 2.2. Method Evaluation of Building Blockchain-Based Supply-Chain System

SCM involves a comprehensive dynamic cooperative relationship among stakeholders, from initial suppliers to end users [23]. Using blockchain technology to reconstruct the traditional supply-chain system is a challenge for developers and managers. Some researchers adopt different analysis and evaluation methods to help relevant practitioners build effective blockchain-based supply-chain systems.

Through a critical examination of blockchain technology and smart contract, Saberi et al. [24] believed that the adoption of blockchain technology in SCM needs to overcome four barriers (inter-organizational, intra-organizational, technical, and external barriers). Considering the uncertainty and emphasis on sustainable transparency, it will be very complicated to select the most appropriate blockchain technology applied in the supply chain. Bai and Sarkis [25] introduced a new hybrid group-decision method, integrated hesitant fuzzy set and regret theory for blockchain-technology selection. Behnke and Janssen [26] investigated four cases in the food supply chain using the interview method and then identified eighteen boundary conditions. Their research implies that taking organizational measures to meet the boundary conditions is a prerequisite for applying blockchain technology in the food-supply-chain system. Fan et al. [27] compared the optimal profits of the supply-chain members in two scenarios (the supply chain adopts the blockchain technology and the supply chain does not adopt it). Analysis and discussion show that not all supply-chain scenarios are suitable for adopting blockchain technology. It depends on traceability awareness and costs.

### 2.3. Blockchain Solutions for the Specific Agri-Food Supply Chain

More efforts have focused on providing blockchain solutions for specific agri-food supply-chain scenarios. Salah et al. [28] leveraged smart contract on Ethereum to control all transactions throughout the supply chain for tracking soybeans. All transactions are recorded in a decentralized ledger linking to an interplanetary file system (IPFS) [29] to provide maximum transparency. Leng et al. [30] proposed a public blockchain for an agricultural supply chain based on a double-chain structure. The double-chain structure can isolate user and transaction information. It also considers the openness and security of transaction information and the privacy of enterprise information. Considering the low scalability and token-driven transaction of public blockchain, some researchers have exploited a consortium-blockchain scheme. Wang et al. [31] proposed a framework based on consortium blockchain to track and trace the workflow of the agri-food supply chain, which is well applied in Shanwei Lvfengyuan Modern Agricultural Development Co., Ltd., Guangdong, China. IoT has the potential to reduce human intervention in the data-collection process and enhance data reliability. Some researchers have studied the combination of IoT and blockchain to establish novel SCM frameworks. Violino et al. [32] combined RFID and blockchain technology to propose a complete electronic traceability prototype to ensure the authenticity of extra-virgin olive oil products, which was applied to a small Italian farm and proved economically sustainable. Tian [33] analyzed the advantages and disadvantages of RFID and blockchain technology in constructing an agri-food supply chain traceability system. On this basis, the researcher showed the process of building a novel agri-food traceability system by combining the two technologies.

There are many deficiencies in current agri-food traceability solutions based on blockchain technology. Articles [28] do not consider the low throughput and gas consumption of the public chain architecture. Taking Ethereum as an example, the current average throughput is 15~17 TPS. The double-chain structure used for privacy isolation and data layering increases storage consumption [30]. This indicates that a simple architecture is needed to protect private data and reduce storage pressure on the chain. These are all what we need to consider in our current work.

In summary, as a branch of the supply chain, the agri-food supply chain should solve the shortcomings of the traditional SCM and also consider the characteristics of the agri-food supply chain. For Xinjiang jujube SCM, it needs to overcome the difficulties of conventional supply-chain-information docking, data easy to tamper with, and a data-storage mechanism prone to single-point failure. RFID technology can improve the efficiency of data docking but also exposes many problems, such as single-point failure, data tampering, and high implementation cost. Blockchain technology shows the potential to overcome the above issues. However, the supply chain of jujube in Xinjiang also has particular problems. Jujube trading, for example, is a buyer’s market. Not only that, we also need to consider more issues when implementing a jujube SCM framework based on blockchain. For example, chain structure, data storage structure, blockchain performance and other issues. Therefore, we propose an optimized blockchain-enabled jujube SCM framework. This framework overcomes the problems of traditional agri-food SCM while improving the performance of the blockchain network. It also helps jujube farmers strengthen their bargaining power.

## 3. Design and Workflow of Blockchain-Enabled Jujube SCM Framework

### 3.1. Application Scenarios

We investigated the local jujube supply chain in Alar (Xinjiang, China), and found that the whole supply chain is composed of five main phases, namely, plantation, threshing ground, processing factory, warehouse, and retail store. Jujube ripens in four to six months. Local farmers pick the ripe fruits and put them in the local threshing ground for drying. Relevant processors purchase jujubes in batches from production cooperatives with unified purchase prices. Due to the opacity of information exchange and the long-term trade inertia, farmers have almost no bargaining power in this case. Jujubes are packaged into boxes and stored in the warehouse after being processed. Finally, jujubes are transported to retail stores in boxes for sale. Based on our investigation, the application scenario of our jujube SCM framework is shown in Figure 1.

In this framework, there is a front-to-back linear information flow and capital flow transmission route. Farmers and production cooperatives record the growth environment, fertilization, pest prevention, threshing ground environment, drying time, initial price, and other information for jujube. The processor is responsible for recording transportation, processing process, storage conditions, wholesale prices, and additional relevant information. As the final link of the supply chain, retailers contact end consumers to record retail information. Consumers or regulators can trace the whole supply chain of jujube through the constant information on the chain. We find that most research on blockchain-enabled agri-food SCM framework can provide reliable traceability and economic sustainability by relying on this linear model. However, this does not help the vulnerable stakeholders in the agri-food supply chain increase their bargaining power. Within our framework, we enhance communication between farmers and retailers. Farmers can choose to sell the roughly processed jujubes directly to retailers. We establish a separate communication route between the two so that farmers can bypass middlemen, avoid monopoly price bullying, or prevent them from changing their trading strategies to reduce prices after analyzing trading data. This kind of private transaction will not affect consumers or regulators to trace the quality of jujube. It just isolates other stakeholders in the jujube supply chain.

### 3.2. Key Concepts of Hyperledger Fabric

In this article, our blockchain-enabled jujube SCM network is based on Hyperledger Fabric. Hyperledger Fabric is an open-source enterprise-level permissioned distributed-ledger-technology (DLT) platform. It has a modular architecture. Smart contracts can be written in multiple languages to encapsulate business logic. Choosing Hyperledger Fabric to build a jujube SCM framework has the following advantages.

(1)Compared with the financial, game, and other application scenarios, the agri-food supply chain involves fewer stakeholders. As a consortium chain, Hyperledger Fabric can easily control the permissions of network entrants, make each transaction reach a consensus quickly, and increase the scalability of the network.(2)Hyperledger-Fabric networks do not need economic incentives to promote the consensus of transactions in the network to be recorded in the distributed ledger, significantly reducing the use cost.(3)Hyperledger Fabric is highly customizable, which is convenient for managers to establish a network topology suitable for application scenarios. It can use multiple languages to encapsulate business logic, enhancing programmability.

In this section, we will describe its key concepts to facilitate relevant researchers’ understanding of the following work.

Certificate Authority (CA)

Hyperledger-Fabric networks determine the access rights of participants with different digital identities to resources through public key infrastructure (PKI) certificates. A certification authority is a unit that issues PKI-based certificates to members and users in the network.

Organization (Org)

Organizations are members of the blockchain network. Stakeholders in the jujube supply chain are abstracted as organizations in the blockchain network for information exchange.

Chaincode

In Hyperledger Fabric, smart contract is called chaincode. Chaincode is an abstraction of business logic. Peer nodes or external client applications of the blockchain use the chaincode to read and write blockchain state variables.

Peer

Peer is the transaction terminal of an organization and the entity unit in the network. It is mainly used to maintain the distributed ledger and run the chaincode container to read and write the ledger. In essence, the organization’s information exchange and business processes are handled through peer nodes. The types of peer nodes include committing, endorsing, leader, and anchor peer.

Ordering Service

A combination of orderer nodes provides the ordering service. The transactions in the blockchain network are ordered into blocks, and then the blocks are distributed to peer nodes to verify the submission. This process is called consensus.

Channel

A channel is a logical ledger in the Hyperledger Fabric network. Organizations in the same channel share one ledger. The Hyperledger Fabric network can have multiple channels, and organizations on different channels cannot access each other’s ledger.

Policy

Policy defines the access rights of different organization members to resources. It also specifies how many organizations need to agree on a proposal to update resources.

Endorsement

Endorsement means that the peer node executes the chaincode and returns a proposal response to the client application. The client application must collect enough endorsements so that it can submit the transaction to the orderer nodes for ordering.

Membership Service Provider (MSP)

MSP is used to define organizations trusted by the network and provides roles and permissions for members in the network by combining with policies.

World State

The world status represents the latest values of all keys in the transaction log after the chaincode is executed.

### 3.3. Network Topology Architecture of Framework

The blockchain-enabled jujube SCM system is built in layers. The bottom layer is the blockchain network, the middle layer is the smart contract, and the top layer is the application that encapsulates the call of the smart contract according to the business requirements. The design combination of the bottom and middle layers constitutes our blockchain-enabled jujube SCM framework, on which users can customize their applications. The blockchain-network-topology architecture is illustrated in Figure 2.

In the jujube supply chain, there are five main stakeholders, namely farmers, logistics, processors, warehouses, and retailers. According to the design pattern of Hyperledger Fabric, different stakeholders are generally abstracted as organizations to join the network for information exchange. This means that six organizations are connected on one channel in the network topology, including five stakeholder organizations and one organization providing ordering services. Considering that these six organizations need to participate in the process of consensus and verification of the entire network, more organizations will cause more resource consumption and lower network performance. We hope to optimize the network topology. Among the five stakeholder organizations, logistics, processing, and warehousing are mainly the responsibility of the processors. In addition, the operation of these three stakeholders on the blockchain follows the order of the supply chain. That is, logistics, processing, and warehousing are in a linear relationship, and do not require the three to be processed online or in parallel simultaneously. So these three stakeholders are abstracted as an organization. It is only necessary to divide the operations of the three with smart contracts. In the optimized network-topology architecture, org1 represents farmers, org2 represents processors, and org3 represents retailers.

We deploy two CAs per organization. One of them is identity CA, and the other is transport layer security (TLS) CA. Identity CA is used to issue identity certificates to organizations as digital identities in the network, while TLS CA is used for TLS communication to encrypt information and protect security. The purpose of deploying two CAs is to improve the degree of decentralization and enhance the network’s robustness.

The peer node is the communication terminal of the organization, so the specific interaction and physical file storage are through the peer node. For example, the distributed ledger of the jujube supply chain is maintained by the peer node. The developers install the chaincode on the peer node to read and write the blockchain. We express the linear supply chain model from farmers to retailers such that all organizations are connected to the same channel. New blocks are constantly generated, which forms a chain structure of increasing blocks on the channel view. This means that all stakeholders share the ledger, enhance the degree of decentralization, isolate the single-point failure, and facilitate communication. There are private transactions between farmers and retailers. They rely on a separate communication route to prevent processors from viewing the details of their private transactions. Real stakeholders in the organization interact with the blockchain network through visual apps or clients. We give our specific design scheme in Section 3.5.

In this network, transactions need to be ordered before being packaged into blocks and distributed to each peer node for verification. The ordering service is completed by the orderer nodes. Like the peer node, the orderer node belongs to the organization. The orderer organization also needs a separate group of CAs to provide certificate services. The ordering service adopts a deterministic consensus algorithm, so the ledger will not diverge like other permissionless blockchain networks.

The client can entrust the peer node to read and write the blockchain network, where consumers or regulators can obtain traceability data.

### 3.4. Workflow of Common Transaction

Transactions are the basic unit of blockchain-ledger records. The relevant behaviors of stakeholders in the jujube supply chain are ultimately permanently preserved in the blockchain in the form of transaction blocks. In this paper, the linear propagation of information flow and capital flow from farmers to processors and finally to retailers is transparent to all stakeholders in the supply chain. We define the behavior of stakeholders writing traceability information on the blockchain in this linear model as a common transaction. In this section, we will explain the workflow of the common transaction and give the key endorsement-policy configuration.

A common transaction workflow in Hyperledger Fabric is shown in Figure 3. It depicts the main functional nodes, components, and workflow details involved in a common transaction in Hyperledger Fabric.

The following are more details of the key steps in a common transaction workflow. The main components in the workflow are listed in Table 2. In Hyperledger Fabric, all nodes participating in the network must obtain corresponding certificates and public and private key (PK&SK) pairs from the certificate authority. Here, we assume that all nodes have obtained the relevant encryption materials.

(1) Transaction proposal initiation: The client node ci first uses the API provided by the corresponding SDK (node, java, go, etc.) to build the transaction proposal and submit it to the endorsement node set $E=\{e_{1},e_{2},\ldots ,e_{j}\}$. It can be described as follows.
$$ci\to E:Proposal={Sign}_{{SK}_{i}}\{{CID}_{i},CNID,CCID,TS,PD<op,md>\},$$
where CIDi represents a client identity, CNID represents the involved channel, CCID represents invoked chaincode, TS represents timestamp, PD represents payload including operation and metadata (md) and ${SK}_{i}$ represents the private key of the client node i. The client node would use its private key to sign the proposal.

(2) Simulative execution and endorsement: The endorsement node $e_{j}$ receives the proposal; it first verifies the proposal according to the following rules, including whether the format of the transaction proposal is correct, whether the transaction has not been submitted before, whether the signature of the client submitting the transaction proposal is valid, and whether the proposer has corresponding execution authority in the channel. After the verification is passed, $e_{j}$ calls the chaincode to simulate and endorse the response. Finally, $e_{j}$ sends the response to $c_{i}$ which is denoted as follows.
$$e_{j}\to c_{i}:Response={Sign}_{{SK}_{j}}\{RespValue,RW\},$$
where RW represents the read-write set and ${SK}_{j}$ represents the private key of the endorsement node $e_{j}$.

(3) Response check and transaction sequencing: Each normal endorsement node of E sends the endorsement result to the client node $c_{i}$. The client node $c_{i}$ checks the endorsement results and sends the transaction to the ordering service O when the endorsement policy is met. Here, we assume that the endorsement policy requires the signature of a peer in organization 1 and a peer in organization 2, which can be denoted as follows:

Policy=AND(Org1.Peer,Org2.Peer). Thus, the message transmission process at this stage can be expressed as follows:$$c_{i}\to O:TxReq=\{CNID,RW,{Sign}_{{SK}_{e1}},{Sign}_{{SK}_{e2}}\}.$$

(4) Validation and update: The ordering service O packages the transactions into blocks and distributes them to all peer nodes in the channel. The peer nodes verify the transactions in the block and update the right set of effective transactions to the world state database.

In the second step, the proposal sent by the client needs to be endorsed and returned to the client after the endorsing peer simulation runs. In the third step, the client must collect enough endorsements according to the endorsement policy before proceeding to the next step. Endorsement is an important part of verifying the transaction and declaring whether the transaction is legal. The endorsement policy we set in the common transaction requires most endorsing peer nodes to endorse, which can be expressed as follows:Policy=MAJORITY(Org1.Peer,Org2.Peer,Org3.Peer).

Such a configuration can make the stakeholders of the supply chain stay online to verify the transaction as much as possible. It does not require all nodes to endorse, which can relatively reduce the time cost.

### 3.5. Design and WorkFlow of Private Transaction

To improve the bargaining power of farmers in the jujube supply chain, we need to build a separate communication route between farmers and retailers in our jujube SCM framework. Farmers can choose to sell the roughly processed jujubes directly to retailers, which ensures that the market determines the purchase price of jujubes and reduces the monopoly price bullying of processors. We define the behavior of farmers and retailers writing information on the blockchain through a separate route as private transactions. Processors cannot know the specific contents of the private transactions, but consumers can trace the contents of private transactions, and regulators can also supervise the contents of private transactions. In this section, we will explore two schemes to realize the private transaction, among which we will choose the scheme suitable for the jujube supply chain. We will also describe the workflow of the private transaction of the optimal scheme and give the key endorsement-policy configuration.

#### 3.5.1. Channel-Level Scheme

Channel can be regarded as a logical ledger. In the channel-level scheme, all stakeholders in the supply chain are abstracted as organizations connected to a channel-sharing ledger. Stakeholder organizations that need to conduct the private transaction are connected to another channel that excludes other organizations. Figure 4 shows the scenario where a channel-level scheme is adopted. All three organizations are connected to channel 1 and share the same ledger. Organization 1 and organization 3 are also connected to channel 2 and conduct the private transaction.

In the channel-level scheme, creating a separate channel will incur additional management overhead (maintaining the chain code version, policies, etc.). It is impossible to allow all channel participants to save the transaction summary while keeping part of the data private. This means that the isolation and fragmentation of distributed ledgers reduce the degree of decentralization and the robustness of the network to a certain extent. The structure of the jujube supply-chain model is relatively simple. The blockchain network formed after abstracting the jujube supply-chain model should not be complicated, which would increase the difficulty of management.

#### 3.5.2. Organization-Level Scheme Using Private Data

Private data are stored separately in a private database on the nodes of the authorized organization. When all organizations are connected to the same channel, private data provides transaction isolation at the organization level. Unauthorized organization nodes can only view the hash value of private data. Hash value is written into the ledger of all peers in the channel after being endorsed and ordered. 

Unlike the channel-level scheme, organizations that conduct private transactions no longer need to connect to a new channel that does not include other organizations. Compared with the channel-level scheme, this scheme reduces the network overhead. Meanwhile, in the organization-level scheme using private data, the workflow of the private transaction is more complex. The transaction workflow is depicted in Figure 5.

The following are more details of the key steps in this transaction workflow. Compared with the common transaction workflow, the transaction workflow using private data requires more components which are listed in Table 3.

(1) Transaction proposal initiation: The client node ci submits a proposal for calling chaincode to the endorsement node set with private data operation permission. Private data or data generating private data are sent to the transient field of the proposal. The content saved in the transient field is temporary and will be deleted later. It can be denoted as follows:$$c_{i}\to E_{auth}:Proposal={Sign}_{{SK}_{i}}\{Transient(Pv\_data||data),\ldots \},E_{auth}\in E.$$

(2) Simulative execution and endorsement: The authorized endorsement node $e_{k}$ simulates the transaction and stores the private data in the transient data store. According to the organization policy of private data, it distributes private data to other authorized peer nodes through gossip [34]. After endorsement, $e_{k}$ sends the proposal response back to $c_{i}$. The response contains an endorsed read-write set including public data, as well as a hash of private data keys and values. Private data are not sent back to $c_{i}$. The response process can be denoted as follows:

$e_{k}\to ci:Response={Sign}_{{SK}_{k}}\{RW[Pub\_data,Hash(Pv\_K\&Pv\_V)],\ldots \}$.

(3) Response check and transaction sequencing: This step is similar to the common transaction process. The transactions with private data hashes are also included in the block.

(4) Validation and update: Authorized peers first hash the data stored in their local transient data store, and then validate it against the hash in the public block. As long as the validation passes, the private data are copied to the private state store. Finally, the private data are deleted from the transient data store. This step can be denoted as follows:Hash(Transient_Data)≠Hash in Pub_Block?Revert:Update.

There are only two organizations involved in private transactions; endorsement policy needs two organizations to ensure security to the greatest extent, which can be expressed as follows:Policy=AND(Org1.Peer,Org2.Peer).

## 4. Design and Implementation of Key Smart Contracts

For blockchain, the storage resources on the chain are precious. For the writing of jujube planting records as an example, the contents to be written include planting environment, climatic conditions, fertilization data, etc., each of which involves a large amount of descriptive data. When designing smart contracts, we only save the most critical information to the blockchain in the form of key–value pairs. For the description information, we design to store the hash value of the description information. The specific description information is stored under the chain. If the regulator needs to judge whether the description information under the chain is tampered with, he can hash the information under the chain and compare it with the hash value on the chain. The application layer customizes the hash function according to business needs. Here we use the same Keccak algorithm as Ethereum, also known as the SHA-3 algorithm. This algorithm calculates the description information to get a 32-byte digest. In smart contracts, we also store the key and the digest in the form of key–value pairs. Next, we will describe the important smart contract algorithms involved in the common transaction and private transaction. The interaction process adopts the identity of organization 1.

### 4.1. Common Transaction Smart Contract

The common transaction process includes reading and writing the world state. Algorithms 1 and 2 give the smart contract paradigm for these two operations.
**Algorithm 1** Process of Reading Jujube Data in the Common Transaction**Input:**MSP from Org1, Context interface of transaction.**Output:** Decoded jujube data array.    1: **if**
MSP is authorized **then**    2:             Create a result iterator ${RS}_{iterator}$ based on the start and end keys.    3:             Declare the closure of the result iterator in advance.    4:             Declare the result storage space SP.    5:             **while**
$({RS}_{iterator}\to Next)\neq NULL$
**do**    6:                       Parse iteration data to obtain available value ζ.    7:                       Add the results to the storage space: ζ ⊳SP.    8:             **end while**    9:             return SP.    10: **else** return false.    11: **end if**

**Algorithm 2** Process of Writing Jujube Data in the Common Transaction**Input:**MSP from Org1, Context interface of transaction, Predefined jujube data key, Hash value of description information, Relevant critical value.**Output:** Error reporting status.    1: **if**
MSP is authorized **then**    2:             Get data from world state according to key: WS⊳ η.    3:             **if** η≠NULL
**then**    4:                     return error.    5:             **else**    6:                     Assemble the information to be written into a structure θ.    7:                     Format the structure: θ ≫ Θ.    8:                     Write the formatted data into the world state: Θ ⊳WS.    9:                     return error(empty).    10:             **end if**    11: **else** return error.    12: **end if**

### 4.2. Private Transaction Smart Contract

The private transaction process also includes reading and writing the world state, but the functional logic is different from the common transaction process. Algorithms 3 and 4 give the smart contract paradigm for these two operations through pseudo code.
**Algorithm 3** Process of Reading Jujube Data in the Private Transaction**Input:**MSP from Org1, Context interface of transaction.**Output:** Decoded jujube private data array.    1: **if**
MSP is authorized **then**    2:             Create a private result iterator PRSiterator based on the start, end keys and private data collection name.    3:             Declare the closure of the private result iterator in advance.    4:             Declare the private result storage space PSP.    5:             **while**
$({PRS}_{iterator}\to Next)\neq NULL$
**do**    6:                       Parse iteration data to obtain available value ξ.    7:                       Add the results to storage space: ξ ⊳PSP.    8:              **end while**    9:             return PSP.    10: **else** return false.    11: **end if**

**Algorithm 4** Process of Writing Jujube Data in the Private Transaction**Input:**MSP from Org1, Context interface of transaction.**Output:** Error reporting status.    1: **if**
MSP is authorized **then**    2:             Get private data from transient: Transient⊳ Δ.    3:             **if** Δ≠NULL
**then**    4:                       Parse the obtained data to get available data Δ ≫ δ.    5:                       Query result ϕ from private data store PSD according to the key of available data.    6:                       **if** ϕ≠NULL
**then**    7:                                 return error.    8:                      **else**    9:                                 Assemble the parsed data into a structure γ.     10:                               Format the structure: γ≫ Γ.     11:                               Write the formatted data into the private data store: Γ⊳PSD.     12:                               return error(empty).    13:                    **end if**    14:             **else** return error.    15:             **end if**    16: **else** return error.    17: **end if**

## 5. Performance Analysis and Discussion

In this section, we use Hyperledger Caliper to test and analyze the performance of the proposed jujube SCM framework. Hyperledger Caliper is a blockchain-performance-benchmark framework that allows users to test different blockchain solutions using custom use cases and obtain a set of performance test results. The blockchain solutions it supports mainly include Hyperledger Besu, Ethereum, Hyperledger Fabric, FISCO BCOS, etc. After running user-defined test cases, performance metrics such as throughput, latency, and resource consumption can be obtained.

Our benchmark tests were performed in the environment of ubuntu 20.04 with 6 CPU cores, 6 GB memory, and 60 GB hard disk. We conducted the benchmark tests on the four affairs: writing in the common transaction; reading in the common transaction; writing in the private transaction; and reading in the private transaction. Each affair had five sets of benchmark tests. These five benchmark tests conducted 1000 transactions at a fixed rate of 50 TPS, 100 TPS, 150 TPS, 200 TPS, and 250 TPS. We simulated that the client interacted with the blockchain network through the entrusted organization 1.

The status of the transaction results is shown in Table 4, Table 5, Table 6 and Table 7. We found that whether writing in the common transaction or private transaction, single-digit failure results occurred at the send rate above 100 TPS. After carefully searching the logs during the benchmark process, we found that failed transactions occurred in the first ten transactions. This is because the rate controller of Hyperledger Caliper has an acceleration process at the beginning, and the send rate is unstable at that time. After the send rate entered a stable period, all transactions were successful. When we tested reading in the common transaction and private transaction, we used random numbers to read the transaction content randomly to simulate the real situation of reading data under traceability or regulation. This caused the number of reading failures to occasionally be a little higher than the number of writing failures. This is the result of the same random number.

The performance metrics of throughput and latency are shown in Figure 6.

In Figure 6, it is obvious that reading records are better than writing records in the performance metrics of throughput and average latency. This is because writing records involves the process of endorsement and consensus, while reading records only requires reading the world state database of the node. In Figure 6a, although the send rate can reach 250 TPS, the throughput limit of writing records in the common transaction is about 78 TPS, and the throughput limit of writing records in the private transaction is about 83 TPS. The throughput of writing records in the private transaction is higher than that in the common transaction. Writing records in the common transaction involves the endorsement and consensus of three organizations, while writing records in the private transaction only involve the endorsement and consensus of two organizations, which is the main reason why the latter is better than the former. This proves that our design in Section 2 is reasonable. That is, logistics, processing, and warehousing are abstracted into one organization rather than three organizations respectively. This reduces the number of organizations participating in endorsement and consensus. Figure 6b shows that the throughput limit of reading records in the common transaction and private transaction is about 220 TPS. When the send rate is 200 TPS, the throughput performance of reading records in the private transaction is significantly lower than that in the common transaction. Thrashing causes this result. In Figure 6c,d, we can see that the average latency of reading and writing records is excellent in different situations.

We visualized the resource consumption of the four main nodes at 250 TPS, the send rate with the best overall throughput performance, and average latency. The performance metrics of resource consumption are shown in Figure 7.

In Figure 7a, peer1 is the node with the highest CPU utilization. When writing records in the common transaction, its CPU utilization is the highest, about 43%. In Figure 7b, the average memory utilization of the three peer nodes is basically the same, ranging from 140 MB to 160 MB. The average memory utilization of the orderer node is low, with a maximum of 60 MB.

Through the benchmark test, it could be seen that our blockchain-enabled jujube SCM framework can be deployed on the host with a general configuration to obtain ideal performance. Table 8 shows the comparison between our research and previous research. We selected three outstanding previous studies.

The RFID solution itself is a centralized architecture. The advantage of centralization lies in its read-write performance. The blockchain solution is a decentralized architecture and requires nodes to reach consensus. Therefore, the read-write performance has a natural disadvantage compared with centralized architecture. In order to improve the read and write performance of the SCM framework, we compressed large data on the chain and optimized the topology of the blockchain network. Through the analysis of the benchmark test report, the throughput and latency were greatly improved compared with other blockchain solutions. The blockchain solution itself has advantages in information sharing, robustness and transparency. For privacy isolation, the double-chain structure increases the complexity of the whole framework and reduces the robustness. This study proposes private transactions to isolate private data by dividing transaction workflows, without affecting the robustness of the framework.

## 6. Conclusions

This paper proposed a framework of jujube SCM based on blockchain. This framework not only realized the reliable traceability of jujube supply-chain data, but also increased the bargaining power of jujube farmers in the supply chain by introducing the workflow of private transactions. The implementation was conducted on Hyperledger Fabric, a distributed ledger-solutions platform. This paper presented an innovative approach to the traditional Hyperledger Fabric underlying network-construction paradigm, which improves the performance of the framework by integrating communication units. The performance metrics of the framework were obtained through Hyperledger Caliper, including throughput, latency, and resource consumption. Through the analysis of relevant results, it was found that reducing the number of nodes participating in endorsement and consensus can improve the performance of the framework, but this needs to be balanced with the degree of decentralization. The stakeholders involved in the jujube supply chain are not complicated. The blockchain-enabled jujube SCM framework proposed in this paper could meet business needs well. Compared with previous studies, it can be seen that the framework has greatly improved the read-write performance. Private transaction workflow, as a novel privacy-isolation strategy, is used to enhance the bargaining power of jujube farmers. It does not degrade the read-write performance of the SCM framework. This gives relevant researchers a new direction to consider in data classification, so as to build a blockchain network with clear structure and no complexity. This helps improve management and performance. This is also conducive to the supervision of relevant supply-chain data by regulators. The improvement of blockchain performance in this study is still limited to large-scale network-topology optimization. We have not conducted a fine-grained study on the impact of consensus algorithm, block size, and block-creation interval on blockchain performance. In future work, we will carry out the above fine-grained research, which could help use blockchain technology to reconstruct a more complex supply chain.

## Figures and Tables

**Figure 1 foods-12-00587-f001:**
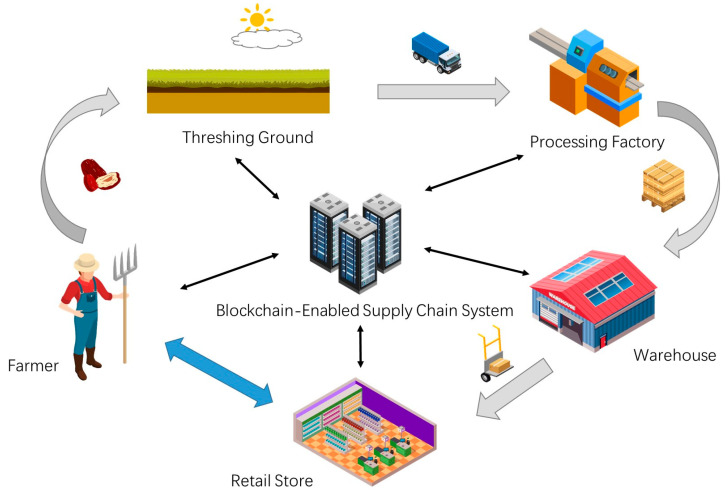
Application scenarios of blockchain-enabled jujube SCM framework.

**Figure 2 foods-12-00587-f002:**
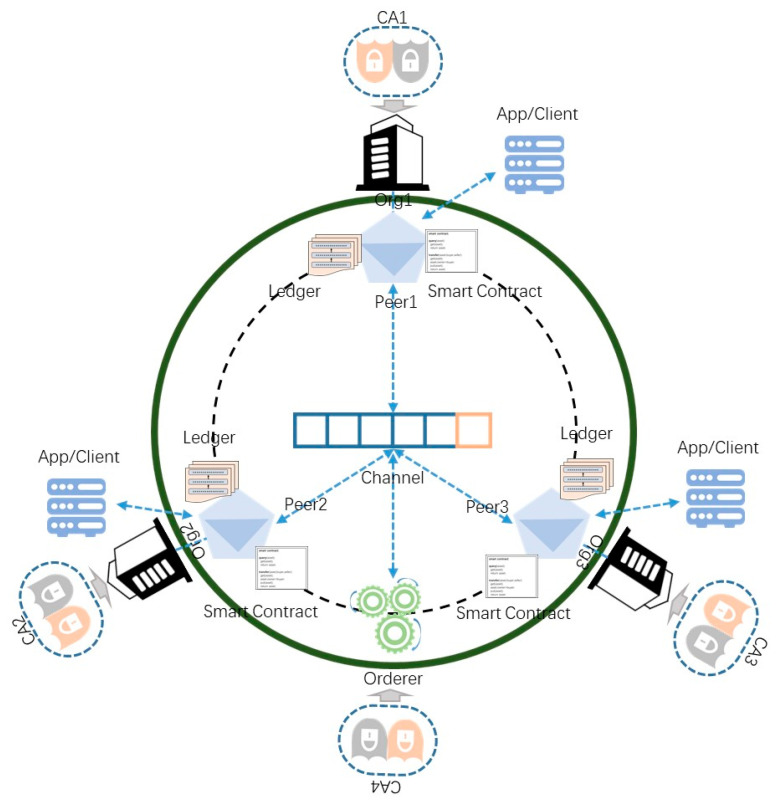
Network-topology architecture.

**Figure 3 foods-12-00587-f003:**
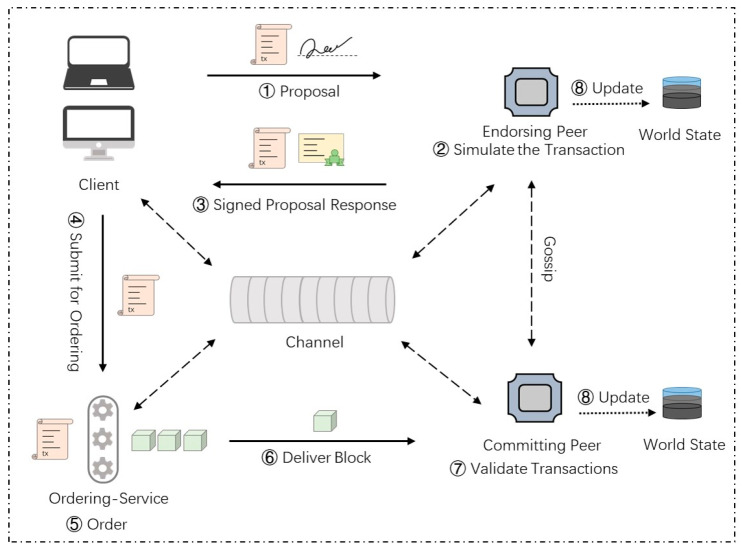
Workflow of a common transaction.

**Figure 4 foods-12-00587-f004:**
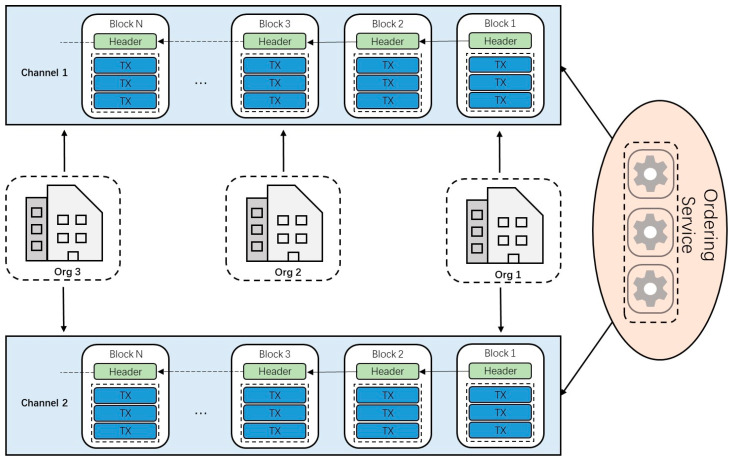
Demonstration of the channel-level scheme.

**Figure 5 foods-12-00587-f005:**
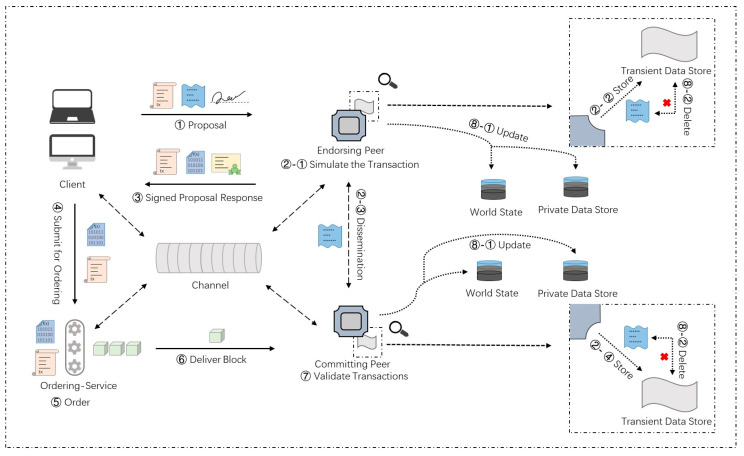
Transaction workflow in the organization-level scheme.

**Figure 6 foods-12-00587-f006:**
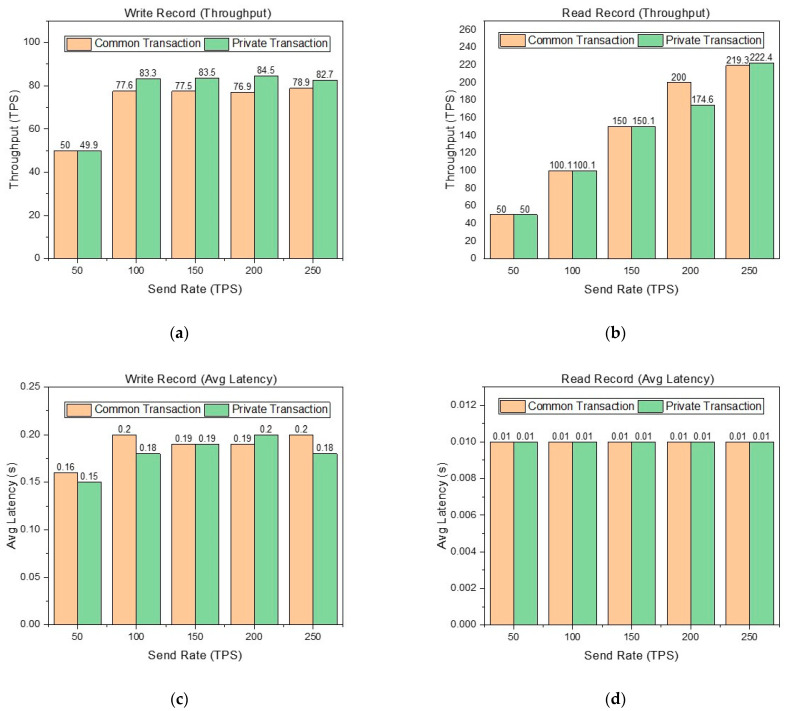
(**a**) Throughput of writing records in the common transaction and private transaction at different sending rates. (**b**) Throughput of reading records in the common transaction and private transaction at different sending rates. (**c**) Average latency of writing records in the common transaction and private transaction at different sending rates. (**d**) Average latency of reading records in the common transaction and private transaction at different sending rates.

**Figure 7 foods-12-00587-f007:**
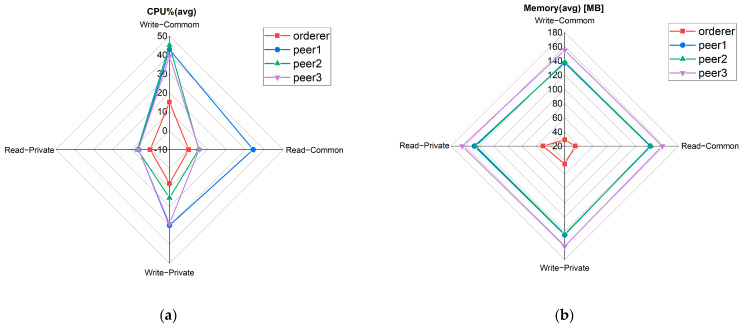
(**a**) The average CPU utilization of the four main nodes to write and read records in the common transaction and private transaction. (**b**) The average memory utilization of the four main nodes to write and read records in the common transaction and private transaction.

**Table 1 foods-12-00587-t001:** Comparison of three different architectures of traceability platforms.

Number	Tampering Cost	Data Backup	Storage Cost	Trusted Environment	Transparency
1	Low	Few	Low	Necessary	Low
2	Low	Few	Medium	Necessary	Medium
3	High	Many	High	Unnecessary	High

**Table 2 foods-12-00587-t002:** Main components in a common transaction workflow.

Components	Description
Client	The initiator of a transaction proposal
Peer	The transaction endpoint of an organization
Orderer	The collection that sorts transactions and packages them into blocks
World State	The component that records the latest values of the transaction log
Channel	The isolation unit for broadcasting information, data, and ledger

**Table 3 foods-12-00587-t003:** More components in the transaction process using private data.

Components	Description
Transient Data Store	The temporary storage area for transient data
Private Data Store	The database dedicated to storing private data

**Table 4 foods-12-00587-t004:** Transaction status of 1000 writing in the common transaction.

Status	Send Rate (50 TPS)	Send Rate (100 TPS)	Send Rate (150 TPS)	Send Rate (200 TPS)	Send Rate (250 TPS)
Success	1000	998	993	993	993
Fail	0	2	7	7	7

**Table 5 foods-12-00587-t005:** Transaction status of 1000 reading in the common transaction.

Status	Send Rate (50TPS)	Send Rate (100 TPS)	Send Rate (150 TPS)	Send Rate (200 TPS)	Send Rate (250 TPS)
Success	1000	1000	995	993	995
Fail	0	0	5	7	5

**Table 6 foods-12-00587-t006:** Transaction status of 1000 writing in the private transaction.

Status	Send Rate (50 TPS)	Send Rate (100 TPS)	Send Rate (150 TPS)	Send Rate (200 TPS)	Send Rate (250 TPS)
Success	1000	998	994	994	994
Fail	0	2	6	6	6

**Table 7 foods-12-00587-t007:** Transaction status of 1000 reading in the private transaction.

Status	Send Rate (50 TPS)	Send Rate (100 TPS)	Send Rate (150 TPS)	Send Rate (200 TPS)	Send Rate (250 TPS)
Success	1000	999	995	992	995
Fail	0	1	5	8	5

**Table 8 foods-12-00587-t008:** Comparison with previous studies.

Study	Architecture	Throughput	Delay	Information Sharing	Robustness	Transparency	Privacy Isolation
RFID solution [21]	Centralization	High	Low	Difficult	Low	Low	
Blockchain solution [28]	Public blockchain	Low	High	Simple	High	High	Low
Blockchain solution [30]	Public blockchain(double chain)	Low	High	Medium	Medium	High	High
Our solution	Consortium blockchain	Medium	Low	Simple	High	High	High

## Data Availability

Data is contained within the article.

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
