# Peer review of "A Novel Blockchain-Enabled Supply-Chain Management Framework for Xinjiang Jujube: Research on Optimized Blockchain Considering Private Transactions"

_foods, 2023, doi:10.3390/foods12030587_

Round 1
Reviewer 1 Report
the paper proposes a Blockchain-Enabled Supply Chain Management Framework for Xinjiang Jujube. All the paper is well-written and organized. however, the introduction can be improved by highlighting the key characteristics of the proposed scheme. Also, figures shall be improved for readabilty.
The results are presented but compatrsion with related work is encouraged to show improatance of the propsoed work
Reviewer 2 Report
Authors propose an optimized blockchain-enabled jujube SCM framework. They contend that this blockchain enabled SCM framework which overcomes the problems of traditional agri-food SCM while helping jujube farmers strengthen their bargaining power.
As the authors wrote (Page 2, line 70-71) “….. current research highlights the characteristics that the agri-food information is difficult to tamper with, and the traceability information is easy to trace and does not take advantage of other merits of blockchain.” I found interesting that Authors were able to develop a blockchain enabled SCM framework ensuring traceability in agri-food supply chain.
It is well written, well designed and organized, and thought-provoking paper; It was a pleasure for me to read this paper. However, if the authors address the following comments, I think they will strengthen their paper.
· When I read the paper, I was able to verify traceability but not sustainability. Could you illustrate how this model meets sustainability challenge?
· In conclusion authors (Page 18, lines 565-567) state that “In future work, the authors will evaluate the impact of blockchain block size, block creation interval, and consensus protocol on the performance of the framework, which could help use blockchain technology to reconstruct a more complex supply chain.” Normally we elaborate what could be done in future, not only what the authors would do in future.
· What are the limitations of the study? How particular this framework for jujube? Can it be used some other agrifood supply chains?
· I was expecting more articles from 2022 since blockchain is such a hot topic. I found 2 of them
1. “Towards user-centered and legally relevant smart-contract development: A systematic literature review”
2. “Assessing blockchain technology adoption in the Norwegian oil and gas industry using Bayesian Best Worst Method.”
Both from “Journal of Industrial Information Integration” I suggest to the authors to review these articles and cite if there is anything that enhances their study. (I understand there are so many articles we cannot cite all of them, but I think we should peruse the ones published in 2022 for currency of our research.)
Some minor points:
· “PKI-based certificates” though PKI is quite well known for Blockchain people I think it is better to be spelled out when it is mentioned first in the paper
· Page 6 line 240 authors wrote “chaincode container” whereas definition of chaincode is in line 248; definition should be first.
· TLS needs to be spelled out when first mentioned though it could be quite common for blockchain community.
· There are 2 figure 6s. The last one is figure 7.
Unfortunately, I could not evaluate algorithms 1 through 4 which I think very important to decide whether to publish this article or not. We need the other reviewers’ evaluations.
Overall, I think that there is enough contribution to literature ---- assuming algorithms 1-4 have no logical error
Reviewer 3 Report
The authors submitted a manuscript that is interesting, and it is linked to the objectives of the journal, however, there are some issues that have to be reconsidered.
The objective of the manuscript is to analyse a proposal of a jujube supply chain management framework based on blockchain.
The topic addressed is relevant for the journal and for the field, as large. The aimed gap is pointed out from studied literature review so, but it is not clear enough, as seems to be more a business gap, than a scientific gap. There is necessary to further arguments to see what is this study bringing new in the scientific literature, but covering a gap that is, at the moment, unexploited and insufficiently exploited by other scholars.
The subject area is rather interesting, so there is potential room for this manuscript, once it reaches the expected level of quality.
The Abstract is too long, also the main results are missing. Line 10 something is missing “Jujube is important economic agri-food in Xinjiang, China”. It is a product, input etc.? The abstract should be the high-level summary of a paper and not a simple enumeration of some points grabbed from the paper. It is advisable to make it simpler and clearer.
For better visibility on databases, the authors are asked not to repeat among keywords the words/concepts included in the title of the article. Entering different words in the title and in the keywords can improve the search for the paper in metasearch engines and internet databases.
In the introduction, the presentation of the structure of the paper is missing. Also, the main objective and the potential secondary objectives are not sufficiently pointed out. Line 30-31 “Jujube is a critical economic agricultural product in Xinjiang, and its jujube output accounts for 37.2% of the global jujube market [1].”, the marked work should be deleted, as it is redundant.
The part of the Literature Review has to be reconsidered, as it is insufficiently developed. There are missing relevant information that are needed to construct the literature gap. General text about the topic (one para), focus on its relevance and importance for the industry, practice, and theory. Then briefly inform readers that there is a surge in wellness experience, and xx and xx studies have been organized so far. One para on what has been done so far on the topic. Then talk about research gaps- What are the keys and why are they important or need to be addressed now? Clearly present 3-4 research gaps on this topic. Finally, present the focus of the current study. What are its RQs and details on the method – briefly say about data, country, context, and theory. Which is the novelty and contribution of the study?
The methodology needs a deeper explanation, so to sounds solid and to be well conceptualized. More details about how the data were acquired should be mentioned as “2.1. Sample selection” section, as it is not clear if the 163 documents selected are all available documents on the subjects, or they were selected based on some specific criteria. Why those criteria are relevant? Why the interval of 2013-2022 was chosen? On the lines 117-119 it is mentioned that community governance policies are analyzed from the perspectives of "value", "purpose", "tool", "value-time", "purpose-time", and "tool-time", while on the sub-chapters 2.2.1.-2.2.3, only 3 of them are detailed (as they are used on Figure 1). What is the reason for value-time", "purpose-time", and "tool-time"? "Value" and "value-time" are not the same perspective, anyway. Some clarifications are needed here
The results. (“3. Design and Workflow of Blockchain-enabled Jujube SCM Framework”).
3.1. Application Scenarios. For better understands of the model, the number (even estimations) of the stakeholders would help (how many farmers/associations, retailers, processors, storages; how big/powerful they are etc.).
Overall, the results and their discussions look, for me, more a business-model than a scientific one, so I suggest to clarify that.
The discussion is almost missing, from scientific part of view. I suggest to show an adequate comparison of the results with the previous literature, however, the authors must emphasize the contribution of the manuscript to the literature, leading to theoretical implications. That means creating an adequate literature review.
The conclusions seem to be superficial, there are missing the limits of the research (which exists in fact to this study) and there are no recommendations for the government, scholars, forests/parks administrators etc.
The references are valid but not consistent as, again, the scientific literature gap is almost missing.
The tables and figures are clear and enough.
Other:
Some phrases are too long (see lines 176-179, but not only). It is better to split them in shorter and clearer sentences.
Round 2
Reviewer 1 Report
Authors have implemented comments in the previous round
Reviewer 3 Report
The authors improved the manuscript. However, there are some minor issues to be fixed for being accepted for publishing, as is:
1. the keywords could be better differentiated by avoiding the repetition of any of the concepts in the title.